# The Apelinergic System: Apelin, ELABELA, and APJ Action on Cell Apoptosis: Anti-Apoptotic or Pro-Apoptotic Effect?

**DOI:** 10.3390/cells12010150

**Published:** 2022-12-30

**Authors:** Natalia Respekta, Karolina Pich, Monika Dawid, Ewa Mlyczyńska, Patrycja Kurowska, Agnieszka Rak

**Affiliations:** Laboratory of Physiology and Toxicology of Reproduction, Institute of Zoology and Biomedical Research, Jagiellonian University in Krakow, 30-387 Krakow, Poland

**Keywords:** apelin, ELABELA, APJ, apelinergic system, apoptosis

## Abstract

The apelinergic system comprises two peptide ligands, apelin and ELABELA, and their cognate G-protein-coupled receptor, the apelin receptor APJ. Apelin is a peptide that was isolated from bovine stomach extracts; the distribution of the four main active forms, apelin-36, -17, -13, and pyr-apelin-13 differs between tissues. The mature form of ELABELA-32 can be transformed into forms called ELABELA-11 or -21. The biological function of the apelinergic system is multifaceted, and includes the regulation of angiogenesis, body fluid homeostasis, energy metabolism, and functioning of the cardiovascular, nervous, respiratory, digestive, and reproductive systems. This review summarises the mechanism of the apelinergic system in cell apoptosis. Depending on the cell/tissue, the apelinergic system modulates cell apoptosis by activating various signalling pathways, including phosphoinositide 3-kinase (PI3K), extracellular signal-regulated protein kinase (ERK1/2), protein kinase B (AKT), 5’AMP-activated protein kinase(AMPK), and protein kinase A (PKA). Apoptosis is critically important during various developmental processes, and any dysfunction leads to pathological conditions such as cancer, autoimmune diseases, and developmental defects. The purpose of this review is to present data that suggest a significant role of the apelinergic system as a potential agent in various therapies.

## 1. Introduction

Carl Vogt described the basis of apoptosis for the first time in 1842, but the term ‘apoptosis’ was only introduced by John Foxton Ross Kerr in 1972; since then, the process has been widely studied in various cells, tissues, and organisms [1]. Apoptosis is a crucial biological process that maintains the homeostasis of organisms, being responsible for cell deletion, development, differentiation, and balancing cell numbers in continuously renewing tissue. The Nomenclature Committee on Cell Death distinguishes intrinsic (also called the mitochondrial pathway) and extrinsic apoptosis. The two pathways are caused by different perturbations: intrinsic by microenvironmental factors such as DNA damage, reactive oxygen species, and mitotic defects, and extrinsic by extracellular factors, for example, tumour necrosis factor-α (TNFα) [2]. B-cell lymphoma 2 family proteins (BCL-2) play a key role in intrinsic apoptosis, controlling the most important step: mitochondrial outer membrane permeabilization [1]. During intrinsic apoptosis, cytochrome *c* binds to apoptotic protease-activating factor 1 (APAF-1), which results in a change in the conformation of this molecule. The altered conformation of APAF-1 allows deoxyadenosine triphosphate (dATP) binding, which is followed by oligomerization of this structure, and apoptosome creation [2]. On the other hand, extrinsic apoptosis involves two types of receptors: death and dependence receptors, which receive signals from the cell’s external environment [2]. The intracellular death domain binds adapter proteins, such as the fas-associated death domain, and all these changes lead to activation of caspase-8 and transduction of signal to effector caspases. Both types of apoptosis involve a caspase cascade, initiating caspases 2, 8, and 9 and then executive caspases such as 3, 7, and 14 [2]. Disruption of cell apoptosis has been linked to a variety of disorders, such as autoimmune and neurodegenerative diseases, diabetes, and cancer [2]. Certain authors report on the expression and role of the apelinergic system in the progression of various types of cancer such as breast [3] and kidney [4], suggesting therapeutic potential in the future [5,6]. Several factors such as hormones and neuropeptides can regulate cell apoptosis. Previous studies have documented that both endogenous ligands of the specific receptor APJ (apelin and ELABELA) have an important effect on intrinsic and extrinsic pathways of apoptosis. This review focuses on recent studies on the role of the apelinergic system in the regulation of apoptosis in various cells.

## 2. Structure and Expression of the Apelinergic System

### 2.1. Apelin

Apelin belongs to the large adipokine family, comprising hormones produced by white adipose tissue cells. Research conducted to date has provided a lot of evidence for its multidirectional action and widespread expression in the human organism. The Tatemoto group discovered apelin in 1998, during the search for a ligand for the orphan APJ receptor. The peptide isolated from bovine stomach extracts was able to bind to the APJ receptor [7,8]. Apelin is encoded by the *APLN* gene, located on the X chromosome at position Xq 25–26, which encodes the 77 amino acid pre-propeptide [9]. Its structure has a strong hydrophobic N-terminal region, which is likely to be a signal sequence that influences the interaction with the receptor, and a C-terminal, which is responsible for its biological activity [7]. The human, bovine, rat, and mouse apelin precursors have 76–95% homology and exist as dimers due to disulphide bridges formed between cysteine residues [9]. Mature forms of apelin do not have bridges and occur in monomer forms [10]. As a result of enzymatic hydrolysis, many active forms of apelin may be formed: apelin-36, apelin-17, apelin-13, and the pyro-glutamylated form, pyr-apelin-13 (Figure 1). The amino acid sequence of these mature forms is relatively conserved between species, demonstrating their important physiological role.

The expression of apelin in humans was noted in various structures of the brain and peripheral tissues, such as the spleen, placenta, lungs, ovary, and stomach, as well as in the heart and testis of rodents [10]. However, the distribution of the apelin isoforms varies: apelin-36 is the most common form in the lungs, uterus, and testes, while pyr-apelin-13 is found in the mammary gland, hypothalamus, and placenta [12,13,14]. In humans, the plasma concentration of apelin is in the range of 3–4 ng/mL [15], and there are three isoforms: apelin-17, apelin-13, and pyr-apelin-13 [16,17]. In adipocytes, the expression of apelin increases during adipogenesis and is regulated by factors such as insulin, growth hormone, and TNFα. Nevertheless, the level of these adipokines depends on nutritional status, the amount of adipose tissue, and the plasma concentration of insulin. As with most proteins from the adipokine family, there is a positive correlation between plasma apelin concentration and body mass index [18]. Therefore, significantly increased levels of apelin have been observed in obese people [19].

### 2.2. ELABELA

ELABELA (synonyms: ELA, APELA, Toddler), a newly discovered ligand of APJ, is encoded by the *APELA* gene [11]. This gene is located on chromosome 4 (4q32.3) and, in humans, contains three exons [20,21], while the ELABELA peptide consists of 54 amino acids [11]. The secreting form of ELABELA has a 22-residue signal sequence at the N-terminal. Interestingly, the last 13 residues are highly conserved between species. The C-terminal of ELABELA is also highly conserved and is crucial for binding to the APJ and Gi protein alpha subunit (Gαi) and the β-arrestin 2 signalling pathways [22]. ELABELA is cleaved to a 32-amino acid mature peptide, ELABELA-32, which may be further transformed into ELABELA-11 or ELABELA-21 [23] (Figure 1). ELABELA-32 and ELABELA-11 differ in their membrane-interactive properties, providing a potential mechanism for distinctive signalling outcomes [24].

ELABELA expression was found in the earliest stages of zebrafish development, in the blastula and gastrula [25]. Moreover, its mRNA was also found in the adult prostate and kidney, as well as in renal distal collecting tubes [23]. Studies have also detected its expression in the human cardiovascular system, lungs, and veins [26]. In mice, expression was detected in the placenta and brain (Gene ID: 100038489, NCBI), and, in humans, mRNA expression was detected in numerous tissues, including the ovary, testis, and digestive system. Nevertheless, the highest expression was noted in kidneys, placenta, prostate, and skin (Gene ID: 100506013, NCBI). What is interesting given its hormonal properties is that ELABELA is secreted from cultured human embryonic stem cells (ESCs) [27]. Its levels in human plasma are around 0.34 ± 0.03 nmol/L [26], with a half-life of a few minutes [28]. ELABELA levels depend mainly on the pathological state of the organism; for example, it is downregulated in the rodent pulmonary arterial hypertension model (PAH) [26].

## 3. Pleiotropic Function of the Apelinergic System

### 3.1. Apelin

Apelin is an important hormone that regulates metabolism (Figure 2). It lowers the concentration of glucose in the serum, increasing its uptake and absorption by muscle cells and adipocytes. It also increases the sensitivity of cells to insulin [18]. Interestingly, insulin stimulates the secretion of apelin by adipose tissue cells, while apelin inhibits insulin secretion [19,29]. Because apelin stimulates glucose transport and acts additively to insulin, it is considered an important metabolic hormone that is also involved in the pathogenesis of insulin resistance and diabetes. Many studies have shown an increased concentration of apelin in patients suffering from both type I and type II diabetes. Krist et al. suggested that it could be a compensatory mechanism devoted directly to reducing insulin resistance as apelin itself exerts a different metabolic effect [30]. Studies on isolated murine adipocytes and the 3T3L1 line also indicated that apelin influences lipid metabolism and has an inhibitory effect on lipolysis [31]. Numerous reports show a significant impact of this hormone on the functioning of the cardiovascular system acting as a mitogenic, chemotactic, and anti-apoptotic agent for endothelial cells including in *Xenopus* embryos [32]. In addition, it lowers blood pressure, increases the contractility of the heart muscle and reduces the risk of atherosclerosis, and thus relieves the symptoms of hypertension [33]. Apelin is also an angiogenic factor in mice and humans; it stimulates the maturation and stabilization of vascular cells in physiological angiogenesis [34], as well as during neoangiogenesis [35], mainly by the induction of proliferation and migration. Additionally, it participates in the neoplastic progression of ovarian cancer cells [36] and the invasive growth of lymphatic microvessels in vivo, which, in the future, may lead to novel anti-cancer therapies targeting lymphangiogenesis [37]. Apelin can also regulate inflammation; hence, it is sometimes classified as a cytokine. In rats, it has the potential to inhibit the secretion of pro-inflammatory mediators, protecting tissues from excessive inflammatory responses [38]. The beneficial effect of apelin also manifests itself in inhibiting the secretion of free radicals in adipose tissue [39]. There is evidence that apelin plays a role in regulating hypothalamic water uptake, thereby contributing to the maintenance of fluid homeostasis [40]. Apelin is also an important regulator of reproductive functions. It influences steroidogenesis in ovarian cells by stimulating the secretion of progesterone and estradiol by increasing the expression of the enzymes 3β hydroxysteroid dehydrogenase and aromatase in humans, pigs, and cattle [41,42,43].

### 3.2. ELABELA

ELABELA can act via APJ and induces different downstream signalling pathways, such as PI3K/AKT/mammalian target of rapamycin kinase (mTOR) [45] and mitogen-activated protein kinase 3 (MAPK3/1) [46], and participates in the regulation of various processes including embryo development, heart, and renal function, food intake, and placenta function (Figure 2). For example, ELABELA-32 increases cardiac contractility in rats and induces coronary vasodilation via MAPK3/1 [47], interestingly being more efficient than apelin [48]. Moreover, via the activation of APJ, ELABELA-32 attenuates the increase in right ventricular systolic pressure, ventricular hypertrophy, and pulmonary vascular remodeling in cardiopulmonary tissues from PAH rat models and patients via the MAPK3/1 pathway [26], while in mice, it exerts a vasodilatory effect on the coronary artery by inhibiting the potentiation of Ang II-induced vasopressor [46]. On the other hand, ELABELA-21 in the paraventricular nucleus leads to elevated blood pressure in rats, and finally hypertension via activation of the PI3K/AKT pathway [49], which indicates an isoform-dependent effect. Interestingly, some studies have suggested an effect of ELABELA-32 on preeclampsia during pregnancy, which is characterised by hypertension, because ELABELA-32 induces the upregulation of matrix metalloproteinase-2 and matrix metallopeptidase-9 (MMP9) expression due to the activation of PI3K/AKT/mTOR, which is strictly connected with invasion and migration associated with early-onset preeclampsia pathogenesis in human HTR8/SVneo cells [46]. Another interesting property of ELABELA is the regulation of food intake, for instance, by suppressing appetite regulation in mice via activation of corticoliberin and arginine vasopressin (AVP) neurons in the hypothalamus [13,50]. Similarly, in adult rats, ELABELA-32 regulates fluid homeostasis by increasing diuresis and water intake by inhibiting vasopressin [46], which is regulated by activation of G_i_ signalling or AVP [50]. Finally, ELABELA/APJ regulates zebrafish skeletal development, bone formation, and bone homeostasis by inhibiting SRY-box transcription factor 32 levels in ventrolateral endodermal cells, indirectly leading to changes in the expression of bone morphogenetic protein targets [51].

## 4. APJ Receptor: Structure and Function

The APJ receptor belongs to class A of the G protein-coupled receptor (GPCR) family and was originally isolated in 1993 [52]. Seven transmembrane domains within GPCR proteins such as APJ provide the site for the processes necessary to maintain normal receptor function: PKA phosphorylation, C-terminal palmitoylation, and N-terminal glycosylation [52]. Individual domains play specific roles in, among others: connecting the receptor with the cell membrane or maintaining its expression and stability (glycosylation) [53]. Until 1998, before the specific ligand called apelin was discovered, APJ was considered an orphan receptor with a high (approximately 50%) homology to angiotensin II [7,52]. APJ consists of 380 amino acids, and its gene (*APLNR*) lies within the long arm of chromosome 11 (q12) [52]. Interestingly, numerous studies have shown that, within different species, the structure of APJ is highly homologous—in humans, mice, and rats the sequence similarity of the receptor is over 90% [10,44,54]. Interestingly, research conducted by Zou et al. (2000) also showed that APJ can act with cluster of differentiation 4 as a coreceptor for human (HIV) and simian immunodeficiency viruses, and it was observed that individual isoforms of apelin block entry of these viruses into cells by inhibiting HIV infection [55]. Moreover, the expression and importance of the receptor has been studied in numerous cells, tissues, and whole animals (e.g., mice, rats, *Xenopus*) and humans, where it regulates several processes, including angiogenesis, body fluid homeostasis, energy metabolism, and functions of the cardiovascular system [10,15,35,56].

APJ expression is affected by numerous substances, including oestrogens, insulin, and stress factors, while the signalling pathway mediates particular cells and processes, and APJ activation occurs via specific ligands, such as individual apelin and ELABELA isoforms [10,11]. The available data show that signal transduction by GPCR receptors is complex, and, in the case of the APJ receptor described above, it interacts differently in specific cells through three different isoforms of G protein, namely Gα, Gβ, and Gγ [7]. It was also reported that in Chinese hamster ovary (CHO) cells, APJ signal transduction influences the phosphorylation of ERK1/2 and AKT and inhibition of cyclic adenosine monophosphate production mediated by Gi protein alpha subunit (Gαi/o) proteins [20]. In addition, it was shown that the aforementioned pertussis toxin-sensitive Gαi/o proteins, as well as those insensitive to it (Gαq/11), participate in apelin signal transmission through the receptor in rat cardiomyocytes, influencing the phospholipase C—protein kinase C pathway and thus stimulating cell contractility [57]. Additionally, depending on the site of action of the apelinergic system in human embryonic kidney cells (HEK), microvascular endothelial cells, and neurons, the internalization of the APJ receptor depends on the ligand isoform. Studies showed that apelin-36 induces this process and binds more strongly with β-arrestin than apelin-13 [8,58]. The notion that individual isoforms of apelin are tissue-dependent is supported by the fact that in HEK cells, apelin-36 and -13 are capable of transporting calcium ions, while in CHO and neurons they do not play such a role [9,59]. Interestingly, apelinergic signalling in human umbilical vein cells also phosphorylates the ribosomal S6 kinase, involving the cascade of the PI3K and ERK1/2 pathway, and thus stimulating the proliferation of these cells [60]. In addition, we found that apelin increases the secretion of placental protein hormones by activating APJ and ERK1/2 kinases and, in the case of steroid hormones, PKA [61]. Both apelin and ELABELA are a ligand for the APJ receptor; although they have many similarities regarding their functions in the body, they act through different signalling pathways and have different biological effects. For example, ELABELA and apelin can antagonise the renin–angiotensin system; apelin upregulates angiotensin, converting enzyme-2 (ACE2) via PKC, and reduces angiotensin II expression, while ELABELA downregulation ACE via PKC-independent mechanisms [44] (Figure 3).

### 4.1. Nervous System

The neuroprotective effect of the apelin/APJ system, including the suppression of apoptosis, has been described previously, and involves many signalling pathways (Table 1). In cultured mouse cortical neurons, treatment with apelin-13 decreases cytochrome *c* release, inactivates caspase-3, and reduces reactive oxygen species (ROS) via the PI3K/AKT signalling pathway [66]. Apelin-36 also acts through this signalling pathway in the neonatal hypoxic/ischemic injury model in rats, demonstrating a neuroprotective effect [67]. Moreover, apelin-13 may protect SH-SY5Y human neuroblastoma cells against damage and apoptosis induced by rotenone, an environmental neurotoxin that contributes to ATP synthesis disorders and consequently mitochondrial dysfunction [68]. Apelin-13 reduces the effect of 1-methyl-4-phenyl pyridine treatment, such as increased apoptosis in SH-SY5Y, by increasing the expression of the phosphorylated form of ERK1/2 [69]. The latest data suggest a distinct neuroprotective role of APJ heterodimers. In studies by Wang et al., APJ heterodimer decreased the level of cleaved caspase-9 and caspase-3 through the guanine nucleotide-binding protein Gαi1 and Gαq3 signalling pathway, at the same time as increasing the number of neurons in the SH-SY5Y cell line. Additionally, the anti-apoptotic effect of apelin-13 in rat vascular dementia models was demonstrated [70]. The results presented by Xu et al. showed that apelin-13 attenuates brain injury following subarachnoid haemorrhage (SAH) in rats. Apelin leads to inhibition of the activating transcription factor 6 (ATF6), which can reduce endoplasmic reticulum stress-mediated apoptosis after SAH in the rat model. In the same model, administration of apelin may relieve symptoms of cerebral oedema and improve neurological problems by suppressing the activation of the ATF6/CAAT enhancer binding proteins homologous protein-CHOP pathway [71]. This was confirmed by another study that showed attenuated SAH-induced apoptosis by exogenous apelin at 150 µg/kg administered to brain ventricles of rats. Apelin leads to an increase in anti-apoptotic protein Bcl-2 and a decrease in pro-apoptotic protein BCL2 associated X (Bax) and caspase-3 through the glucagon-like peptide-1 receptor/PI3K/AKT signalling pathway [72]. In the case of intracerebral haemorrhage (ICH), apelin has an anti-apoptotic effect by reducing the expression of caspase-3 and upregulating the level of Bcl-2. Moreover, it minimises brain oedema by decreasing the expression of aquaporin-4 and MMP9 and improving motor functions in ICH in mice [73]. Apelin plays a key role in cerebral ischaemia/reperfusion, (I/R) including in the brains of middle cerebral artery occlusion/reperfusion models in rats [74,75] and the transient cerebral I/R model in mice [76,77]. Apelin acts via various signalling pathways affecting neuronal apoptosis, e.g., by Gαi/Gαq-CK2-dependent inhibition of eukaryotic initiation factor 2-ATF4-CHOP activation [74]. Multiple studies have shown the protective effect of apelin against cell apoptosis while allowing for quicker recovery after stroke. Zhu et al. examined the mechanism of action of apelin-36 against neurotoxicity in consequent action of 1-methyl-4-phenyl-1,2,3,6- tetrahydropyridin (MPTP), which induces Parkinson’s disease (PD) in the mouse model. Groups of mice were treated with 0.1–0.5 µg apelin for 7 days, sacrificed, and a series of tests conducted. Apelin leads to the inactivation of caspase-3 and inhibition of the apoptosis signal-regulating kinase 1 (ASK1)/c-Jun N-terminal kinase (JNK) signalling pathway. In addition, apelin increases the expression of *LC3-II* and *Beclin-1* autophagy genes and alleviates oxidative stress in mice treated with MPTP, which may be a novel strategy for PD treatment [78]. Furthermore, apelin increases cell viability and reduces the rate of apoptosis in PC12 rat culture cells, which are a rat model of neuronal cells for neurobiological research [79,80]. Foroughi et al. showed that apelin-13 prevented methamphetamine-induced neurotoxicity; this substance increases the risk of PD [79] and corticosterone, which induces cellular damage [80]. Alzheimer’s disease is another neurodegenerative disease that is associated with amyloid β (Aβ) aggregation and neural loss. Apelin protects against the destructive effects of Aβ on working and spatial memory. Furthermore, apelin decreases caspase-3 protein expression probably via the mTOR signalling pathway in rats [81]. Further research on the efficiency of apelin showed that the neuroprotective mechanism of action includes suppressing apoptosis of hippocampal neurons by down-regulation of Bax and caspase-3 expression in both in vitro and in vivo epilepsy models. Interestingly, experiments have confirmed that apelin is a target gene for miR-182. Increasing the expression of miR-182 may weaken the protective effect of apelin on neuronal damage in the rat epilepsy model [82]. In line with this, apelin is a highly potent neuroprotective agent and should be further explored as a potential treatment under pathological conditions, such as brain injuries, and also Parkinson’s and Alzheimer’s disease.

### 4.2. Cardiovascular System

The apelin/APJ system is abundantly expressed in the cardiovascular system. APJ receptor expression has been demonstrated in the myocardium after I/R injury. Importantly, apoptosis is a major cause of ischaemic heart dysfunction. In cardiomyocytes, apelin has an anti-apoptotic effect through the PI3K/AKT and ERK1/2 signalling pathways. Primary culture of rat myocardial cells confirmed that 30 pmol apelin influences cell viability, inhibits myocardial apoptosis, and prevents resisting oxidation effects [83]. Furthermore, Ouyang et al. showed that pyr-apelin-13 at 200 µg/kg improved mechanical efficiency by inhibiting cardiac fibrosis and apoptosis in the LV (left) myocardium in the rat heart failure model [85]. Moreover, apelin-13 decreases the expression of pro-apoptotic proteins by activating the PI3K/AKT pathway, thus protecting against ischaemic cardiomyocyte apoptosis in rats induced by glucose deprivation [84]. Ustunela et al. examined the potential impact of apelin on a variety of stress-related heart diseases and showed that water immersion and restraint stress led to increased levels of apelin, which inhibited stress-induced apoptosis in rat hearts [86]. Apelin attenuates ER stress-induced apoptosis, reducing CHOP and JNK via PI3K/AKT, MAPK3/1, and endothelial nitric oxide synthase in the rat I/R model [87]. Notably, the apelin/APJ system promotes the expression of myocardial infarction (MI) indicator protein, which leads to increased angiogenesis and improves cardiac repair post-MI by inhibiting apoptosis [88]. Boal et al. determined the effect of apelin on the relationship between I/R and obesity. Interestingly, apelin treatment led to increased levels of phosphorylated forkhead box protein O1 protein, and significantly reduced infarct size in mice fed a high-fat diet and then subjected to cardiac I/R, post-reperfusion. Furthermore, it decreased myocardial apoptosis and mitochondrial damage in a cell line from rat heart tissue, H9C2 [89]. Apelin has a suppressive effect on endothelial apoptosis via the AMPK pathway in human endothelial cells induced by methylglyoxal, which is a glycolytic metabolite detrimental to endothelial dysfunction [90]. It is known that in human vascular smooth muscle cells (VSMCs) apoptosis regulates the remodelling of blood vessels during cardiovascular disease. Apelin-13 administered at 10 pM-1 nM inhibits the expression of the pro-apoptotic protein Bax but increases the expression of the anti-apoptotic protein Bcl-2 by APJ and induces the APJ/PI3K/AKT signalling pathway in VSMCs induced by serum deprivation [106].

### 4.3. Haematopoietic System

Recent studies have shown that bone marrow-derived mesenchymal stem cells have many beneficial functions, including ischaemic tissue repair. Unfortunately, the low viability of bone marrow stromal cells (BMSCs) limits their therapeutic potential. Zeng et al. demonstrated that apelin protects rat BMSCs from apoptotic death because it reduces dose-dependent cytochrome *c* release and caspase-3 activation. The anti-apoptotic effect of apelin is mediated by ERK1/2 and PI3K/AKT signalling. In summary, in the future, apelin may have the therapeutic potential of stem cell therapy for ischaemia-related disorders such as heart ischaemia and ischaemic stroke [65].

### 4.4. Skeletal System

An important role of apelin is its action on bone-building cells through the APJ receptor. Apelin administered at 0.4–10 nM decreased the expression of apoptotic mediators such as caspase-8, caspase-9, caspase-3, and cytochrome *c*, which plays a crucial role in the intrinsic pathway of apoptosis indirectly by forming the apoptosome. In the MC3T3-E1 mouse osteoblastic line, apelin inhibits apoptosis induced by serum deprivation and by glucocorticoid dexamethasone or TNFα. These actions are mediated via the JNK and PI3K/AKT signalling pathways [93]. In human osteoblast-like cells, apelin has a similar effect. Under the same conditions as in the previously described study, apelin reduces the expression of proteins such as Bax, cytochrome *c*, and caspase-3, and, in addition, apelin increases the expression of anti-apoptotic protein Bcl-2. Apelin acts through the APJ/PI3K/AKT signalling pathway. Taken together, apelin protects against apoptosis induced by serum deprivation and induced by the glucocorticoid dexamethasone [91].

### 4.5. Respiratory System

The anti-apoptotic effect of apelin can also be observed in the tissues of the respiratory system, more specifically in the lungs. Firstly, apelin reduces the level of proinflammatory cytokines such as tumour necrosis factor-α and interleukin-6 and smooths inflammation and lipopolysaccharide (LPS)-induced acute lung injury in rats. Secondly, apelin treatment inhibits apoptosis in rat lung tissues and the Beas-2B human lung epithelial cell line. Moreover, overexpression of ASK1 reverses the inhibitory effects of apelin-36 on LPS-induced inflammation and apoptosis. This suggests that its effect may depend on the inhibition of the ASK1/MAPK signalling pathway [94]. Interestingly, Zhang et al. reported an indirect effect of apelin in pulmonary fibrosis. Examination of the effects of melatonin (MLN) on acute lung injury showed that MLN promotes apelin expression. Apelin administered at 100 nM has an anti-apoptotic effect. Moreover, apelin inhibits the production of ROS and increases mitochondrial ATP content, improving the function of mitochondria and protecting against damage in the TC-1 cell line [95].

### 4.6. Digestive System

The cellular immunolocalization method has shown the presence of an APJ/apelin system in cells of the digestive system, which may suggest the regulation of many physiological processes, including apoptosis by apelin. Antushevich et al. observed different effects of apelin depending on the cell line. In the IEC-6 cell line, it decreased the expression of caspase-3, which is a key enzyme responsible for the initiation of the apoptotic process [96]. Other studies by Antushevich et al. showed that administering 100 nmol/kg/day of apelin intraperitoneally to rats for 10 days led to the suppression of apoptosis in tissues of the digestive system. Additionally, caspase-3 expression was reduced in the pancreas and colon [97].

### 4.7. Reproductive System

There are few data showing an unambiguous effect of the apelin/APJ system on apoptosis of reproductive cells. Several studies showed the expression of the APJ receptor and apelin, including in ovarian follicle cells [42,43], where it affects the follicle selection process. Apelin causes a dose-dependent increase (0.02, 0.2, 2, and 20 ng/mL) in the proliferation in porcine granulosa and theca cell cultures. Moreover, apelin at a dose of 2 ng/mL increases the phosphorylation of ERK1/2 and AKT kinases, which promotes cell survival. Using inhibitors of these kinases, PD098059 and LY294002, and the apelin inhibitor ML221, led to the reversal of apelin-stimulated cell proliferation in porcine ovarian follicles [42]. Furthermore, apelin may inhibit apoptosis via the PI3K/AKT signalling pathway in granulosa cells from PCOS rats [43]. The process of apoptosis is important in the development of the transitional organ of the fetus, the placenta. Interestingly, Mlyczyńska et al. showed that apelin increased the mRNA and protein expression of factors inhibiting apoptosis, including baculoviral IAP repeat containing, induced myeloid leukaemia cell differentiation protein and decreased the expression of pro-apoptotic factors, including Bcl-2 and caspase-3, both in the placental cell line (BeWo) and in human villus explants [64]. Moreover, we have also proved the positive effect of the apelinergic system on the proliferation of trophoblast cells through ERK1/2, AKT, AMPK kinases, and signal transducer and activator of signal transducer and activator of transcription 3 [63]. Moreover, studies have shown the participation of the APJ receptor and ERK1/2 and AKT kinases in the anti-apoptotic effect of apelin in human placental cells [63,64].

### 4.8. Urinary System

It turns out that plasma apelin concentrations decrease with age and that the function of many organs, including the kidneys, deteriorates. Interestingly, apelin has a beneficial effect on kidney function in aging mice. Apelin likely influences the physiology of aging, including via the suppression of apoptosis. However, this requires further research [99]. Müller et al. examined the special role of apelin in cardiovascular and diabetic kidney disease (DKD) in mice and demonstrated the presence of the APJ receptor in the glomerular podocytes. Furthermore, apelin reduces caspase-3 activity and abolishes the pro-apoptotic effect of high glucose in hyperglycaemia, suggesting the therapeutic effect of apelin in DKD [100].

### 4.9. Other Tissues and Cells

Apelin also plays a crucial role in other cells, tissues, and organs. For example, apelin could be a potential therapeutic factor for noise-induced hearing loss. Khoshirat et al. demonstrated that apelin inhibits intrinsic apoptosis through the growth of Sirt-1 in cochlear tissue, and, as a result, it decreases caspase-3 expression and the Bax/Bcl-2 ratio, which leads to an improvement in hearing function in rats [102]. Yin et al. observed the anti-apoptotic effect of apelin in mouse HC explants and the mouse auditory cell line HEI-OC1. The TUNEL method showed a decrease in the percentage of apoptotic cells after exposure to apelin in cisplatin-treated cell models. Apelin protects against the toxicity of cisplatin in the inner ear [103]. In contrast, apelin influences both proliferation, migration, and cell survival to reduce apoptosis in rabbit retinal Müller culture cells [104] and rat retinal pericyte cells [105].

## 5. Anti-Apoptotic Effect of ELABELA

In addition to its previously mentioned functions, ELABELA suppresses apoptosis. Recent studies showed that inhibition of APJ signalling by knockout of ELABELA increased apoptosis in the placenta of mice [101]. Moreover, ELABELA contributes to the increased proliferation of BeWo cells [98]. In contrast, other studies have shown that ELABELA promotes cell cycle progression through PI3K/AKT in human ESCs. Additionally, it supports human ESC self-renewal by preventing stress-induced apoptosis [27]. Administration of 5 µM doses of ELABELA to H/I-induced mesenchymal stem cells (MSCs) led to reduced caspase-3 activity and mitochondrial damage. ELABELA-32 also improves rat MSC viability, via the PI3K/AKT and MAP3/1 pathways, enhances BCL2/BAX expression, and decreases caspase 3 activity [92]. ELABELA was also found in rat aortic adventitial fibroblasts in response to angiotensin II. This demonstrates not only the anti-apoptotic role of ELABELA, but also its anti-inflammatory and antioxidant effects via the activation of fibroblast growth factor 21-angiotensin-converting enzyme 2 signalling [62]. Zhang et al. created an in vivo model of type 1 diabetes mellitus mice that showed its anti-apoptotic effect in podocytes and the nephroprotective effect against diabetic nephropathy. ELABELA acts via the PI3K/AKT/mTOR signalling pathway [45]. ELABELA thus shows therapeutic promise.

## 6. Pro-Apoptotic Effect of Apelinergic System

A few studies have indicated a pro-apoptotic effect of the apelinergic system (Table 2). Interestingly, Li et al. showed that ELABELA enhanced p53 expression in murine ESCs, leading to DNA damage-induced apoptosis [107]. Shimizu et al. (2009) showed lutropin-induced expression of apelin and APJ receptors in bovine theca cells, while only the APJ receptor was expressed in bovine granulosa cells. Moreover, the data suggest that apelin has a pro-apoptotic effect because its induction leads to increased mRNA APJ receptor expression in granulosa cells. As a result, the apelin/APJ system may be involved in follicular atresia [108]. In contrast, after 48 h of incubation, apelin had a stimulating effect on the gene expression of *caspase-3* in Caco-2 cells. On the basis of many scientific studies, it has been suggested that apelin impacts the Bax/Bcl-2 ratio [96,102,103]. However, in rat stomach and jejunum cells, apelin increases the expression of caspase-3, which is associated with DNA damage [43]. This suggests that differences in the expression of the apoptosis protein may result from other factors regulating, for instance, the regeneration of the epithelium of tissues and organs in the digestive system.

## 7. Conclusions

In summary, expression of apelinergic system was described in many tissues indicating the involvement in numerous physiological and pathological processes; both apelin and ELABELA at different doses and in various cells regulate apoptosis by activating several kinases such as the PI3K/AKT and ERK1/2 signalling pathways, changing the expression of genes and proteins that mediate apoptosis. Depending on the species/animal model, apelin may have anti-apoptotic effects on the granulosa cells in rats [43], with pro-apoptotic effects in bovine [108]. Depending on the type of cells and doses used, apelin shows different effects, which is the basis for the contradictory data in the literature. Nevertheless, apelin mainly shows anti-apoptotic activity, which is an introduction to research and a potential therapeutic target for various cancers [3,4] but also for many dysfunctions and diseases of the neurological, cardiovascular, and urinary systems (Figure 4).

A second ligand for the APJ receptor was recently discovered, which also regulates cell apoptosis. ELABELA inhibits apoptosis in cardiomyocytes [109] and podocytes [45] via the PI3K/AKT signalling pathway, and also has an effect in BeWo cells [98] and adventitial fibroblasts [62] and in the diabetic model [109]. In addition, studies indicate that both apelin and ELABELA may reduce stem cell apoptosis, which may be a therapeutic target for example ischaemia-related disorders [65]. A detailed description of the mechanisms of apoptosis provided in this review (Table 1 and Table 2) is crucial and helps to understand the pathogenesis of diseases resulting from dysfunctional apoptosis. In turn, this could help develop new drugs targeting specific apoptotic pathways or genes. Perhaps apelin and ELABELA will be new therapeutic targets.

## Figures and Tables

**Figure 1 cells-12-00150-f001:**
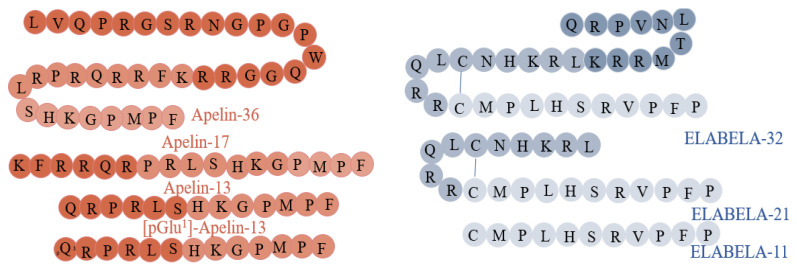
Structure and isoforms of apelin [9] and ELABELA [11].

**Figure 2 cells-12-00150-f002:**
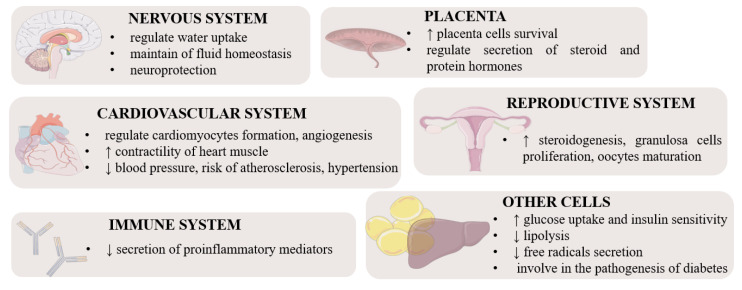
Pleiotropic function of apelin and ELABELA in different tissues [44]. Stimulation, ↑; inhibition, ↓.

**Figure 3 cells-12-00150-f003:**
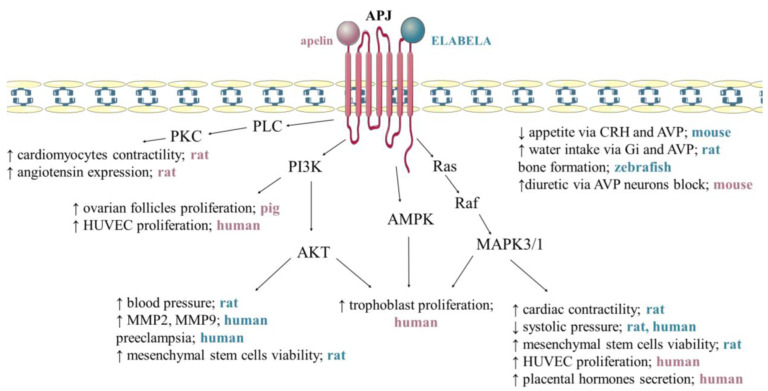
Mechanism of apelin and ELABELA action via APJ receptor. Apelin activates phospholipase C/protein kinase C (PLC/PKC) signalling pathway, increasing cardiomyocyte contractility [57] and angiotensin expression [62]. Apelin stimulates the proliferation of ovarian follicles [42], trophoblast [63], and human umbilical vein endothelial cells (HUVEC) [60] via phosphoinositide 3-kinase/protein kinase B (PI3K/AKT), Raf kinase/serine-threonine-specific protein kinases/mitogen-activated protein kinases (Ras/Raf/MAPK3/1) and 5′AMP-activated protein kinase (AMPK), additionally increases placental hormones secretion [64] and shows diuretic via AVP neurons block [13]. ELABELA increases blood pressure [49], mesenchymal stem cells viability [65], matrix metalloproteinase-2 (MMP2), matrix metallopeptidase-9 (MMP9) [46] via PI3K/AKT. ELABELA activates kinases MAPK/1/3 increases cardiac contractility [47] and decreases systolic pressure [26,46], additionally reduces appetite via cortycoliberin (CRH) and AVP (arginine vasopressin) [46], increases water intake via Gi and AVP [50] and affects bone formation [13]. Stimulation, ↑; inhibition, ↓. The pink lettering refers to apelin. The blue text refers to ELABELA. 5. Anti-Apoptotic Effect of Apelin.

**Figure 4 cells-12-00150-f004:**
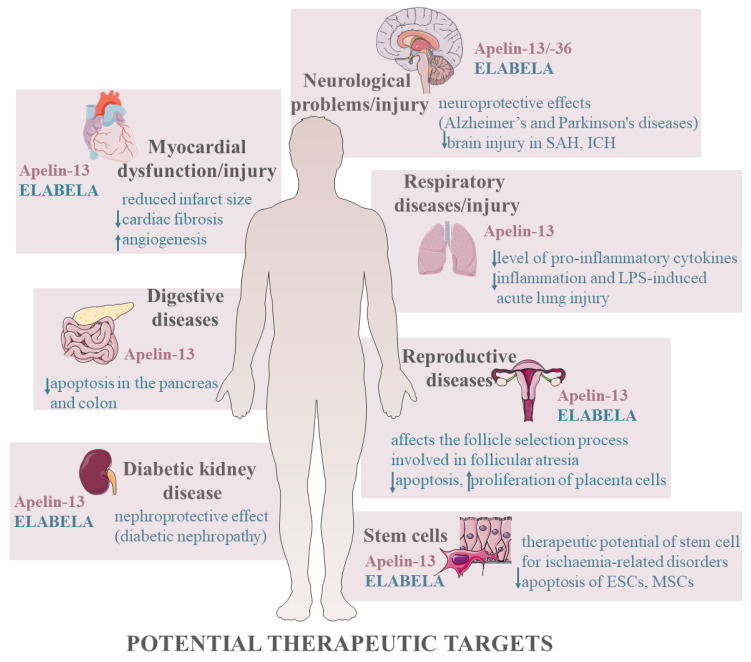
Potential therapeutic targets of apelinergic system. Stimulation, ↑; inhibition, ↓. Abbreviations: SAH, subarachnoid haemorrhage; ICH, intracerebral haemorrhage; LPS, lipopolysaccharide; ESCs, embryonic stem cells; MSCs, mesenchymal stem cells.

**Table 1 cells-12-00150-t001:** Anti-apoptotic effects of apelinergic system. Stimulation, ↑; inhibition, ↓.

Tissues	Species	Doses	Times	Signalling Pathways	References
cortical neurons	mouse	1–5 nM	24 h	↓ROS, ↓cytochrome c, ↓caspase-3, PI3/AKT, ERK1/2	[66]
SH-SY5Y	human	10–9 M	24 h	↓caspase-3 activity	[68]
SH-SY5Y	human	100 nM	2 h	↑p ERK1/2, ↓GRP78/CHOP/cleaved caspase 12	[69]
SH-SY5Y	human	100 nM	2, 4, 16 h	↓caspase-3, ↓caspase-9, Gαi3, Gαq	[70]
PC12	rat	4 µM	1 h prior to METH	↓apoptotic cells, ↓ROS	[79]
PC12	rat	4 µM	0.5 h prior to corticosterone	↓caspase-3, PI3K/AKT, ERK1/2	[80]
hippocampus	rat	2 µg	once a day for 14 days	↓caspase-3, mTOR	[81]
sections of brain	mouse	50 µg	24 h after ICH	↑Bcl-2, ↓Bax, ↓caspase-3,GLP-1R/PI3K/AKT	[72]
hippocampal neurons	rat	overexpression of APLN	48 h post-transfection	↑Bcl-2, ↓Bax, ↓caspase-3, ↓mGluR1, ↑p-AKT, miR-182	[82]
brain	mouse	1 µg	1 h	↓Bax, ↓caspase-3, PI3K/AKT	[67]
brain	rat	50, 150 μg/kg	1 h after SAH	↓ATF6/CHOP, ↓caspase-3, ↑Bcl-2/Bax	[71]
brain	rat	150 µg/kg	0.5 h after SAH	↑Bcl-2, ↓Bax, ↓caspase-3, GLP-1R/PI3K/AKT	[72]
brain	mouse	50 µg	immediately after ICH	↓caspase-3, ↑Bcl-2	[73]
brain	rat	0.5 µg/g	30 min before I/R	↓CK2, ↓eIF2-ATF4-CHOP	[74]
brain	rat	50, 100 μg/kg	immediately after I/R	↓caspase-3	[75]
brain	mouse	50, 100 μg/kg	15 min before I/R	↓caspase-3, ↑Bcl-2, ↓Bax, PI3K/AKT, ERK1/2	[76]
brain	mouse	100 μg/kg	15 min before I/R	↓caspase-3, ↓Bax, ↑Bcl-2, ↑p-AMPK	[77]
brain	mouse	0.5 µg/mice/day	once a day for 7 days	↓cleaved caspase-3, ↓ASK1/JNK	[78]
myocardial cells	rat	30 pmol/L	30 min before hypoxia	↓ROS, ↓apoptotic cells, PI3K/AKT and ERK1/2	[83]
cardiomyocytes	rat	10, 100 nmol/L	12 h	↓caspase-3, ↓Bax, ↑Bcl-2, PI3K/AKT	[84]
cardiomyocytes	mouse	0.1 μg/kg	24 h after I/R	↑Bcl-2, ↓Bax, caspase-3	[73]
heart	rat	200 μg/kg	once a day for 12 weeks	inhibiting apoptosis	[85]
heart	rat	endogenous apelin	-	↑Bcl-2, ↓Bax	[86]
heart	rat	1 μg/kg	15 min before reperfusion	↓caspase-12, ↓CHOP, ↓JNK, PI3K/AKT, ERK1/2, eNOS	[87]
heart	mouse	1 mg/kg	for 14 days after MI	↓number of apoptotic cells, ↑p-AKT, ↑eNOS	[88]
H9C2	rat	1,10,100 nM	24 h	FoxO1	[89]
endothelial cells	mouse	1 μM	24 h	AMPK, CHOP, ROS	[90]
aortic adventitial fibroblasts	rat	100 nM	1h	↓caspase-3, FGF21-ACE2	[62]
		**(ELABELA)**			
VSMCs	human	10 pM–1 nM	48 h	↑Bcl2/Bax, APJ/PI3-K/AKT	[91]
BMSCs	rat	0.1–5 nM	36 h	↓cytochrome c, ↓caspase-3, MAPK/ERK1/2, PI3K/AKT	[65]
MSCs	rat	5 μM	2 h	↑Bcl-2, ↓Bax, ↓caspase-3, ERK1/2, PI3K/AKT	[92]
		**(ELABELA)**			
MC3T3-E1	mouse	0.4–10 nM	48 h	↓cytochrome c, ↓caspase-3, -8, -9, JNK, PI3K/AKT	[93]
human osteoblast-like cells	human	0.4–10 nM	48 h	↓Bax, ↓cytochrome c, ↓caspase-3, ↑Bcl-2, APJ/PI3/AKT	[91]
lungs	rat	10 nmol/kg	24 h	inhibition of the ASK1/MAPK	[94]
Beas-2B	human	1 μM	1 h	inhibition of the ASK1/MAPK	[94]
TC-1	mouse	100 nM	24 h	↓apoptotic cells, ↓ROS, ↑ATP	[95]
IEC-6	rat	10^–8^ M	4–48 h	↓caspase-3	[96]
pancreas and colon epithelial cells	rat	100 nmol/kg/day	for 10 days	↓caspase-3	[97]
ovarian follicles	porcine	2 ng/mL	15 min	↑AKT/ ERK1/2	[42]
granulosa cells	rat	10^−8^ M	12 h	PI3K/AKT	[43]
villous explants	human	2 and 20 ng/mL	24 h, 48 h, 72 h	↓caspase-3, ↑Bcl2/Bax	[64]
BeWo	human	0.5–50 nmol/L	0, 12, 24 h	↑ proliferation cells	[98]
		**(ELABELA)**			
BeWo	human	2 and 20 ng/mL	24 h, 48 h, 72 h	↑XIAP, BCL3, ↓BAK1, BAX, BOK, NOD1, CRADD, caspase-14, -8, -9, -3, -2, APAF1,BAK, BAX, p53, ERK1/2/MAPK, AKT	[63]
podocytes	mouse	100 nmol	24 h	↓caspase-3	[99]
podocytes	mouse	4.5 mg/kg	twice a days	PI3K /AKT/ mTOR	[100]
ESCs	human	-	-	PI3K/AKT	[101]
		**(ELABELA)**			
HCs	rat	100 μg/kg	6 days after noise exposure	↓caspase-3, Bax/Bcl-2	[102]
explants cochlear hair	mouse	10 nM	2 h	↓Bax/Bcl-2	[103]
retinal Müller cells	rabbit	100 ng/mL	12 h	↓caspase-3, ↑Bcl2/Bax	[104]
retinal pericyte cells	rat	100 ng/mL	2 h	↓apoptotic cells	[105]
HEI-OC1	mouse	10 nM	2h	↓caspase-3	[103]

Abbreviations: -, lack of data; h, hours; ↑, increasing; ↓, decreasing; PC12, cell line from a pheochromocytoma of rat adrenal medulla; SH-SY5Y, human neuroblastoma cell line; H9C2, cell line from embryonic BD1X rat heart tissue; APLN, apelin; METH, methamphetamine-induced; ICH, intracerebral haemorrhage; SAH, subarachnoid haemorrhage; I/R, ischaemia/reperfusion; BMSCs, bone mesenchymal stem cells; ROS, reactive oxygen species; MC3T3-E1, osteoblast precursor cell line from mouse 99072810; Beas-2B, human non-tumorigenic lung epithelial cell line from human lung tissue; TC-1, cell line from mouse lung; IEC-6, intestinal epithelial cell line; BeWo, human placental cell line; ESCs, embryonic stem cells; VSMCs, vascular smooth muscle cells; HCs cochlear hair; HEI-OC1 cell line from mouse organ, hair cell-like properties; ASK1, apoptosis signal-regulating kinase 1; JNK, c-Jun N-terminal kinase; GLP-1R, glucagon-like peptide-1 receptor; PI3K, phosphoinositide 3-kinase; AKT, protein kinase B; ERK, extracellular signal-regulated kinase; ATF6, activating transcription factor 6; CHOP, CCAAT-enhancer-binding protein homologous protein; GRP78, glucose-regulated protein 78; Gαi3 and Gαq-containing heterotrimeric G proteins; eIF2, phosphorylation of eukaryotic initiation factor-2; ATF4, activating transcription factor 4; AMPK, 5′AMP-activated protein kinase; MAPK, mitogen-activated protein kinases; eNOS, endothelial nitric oxide synthase; FoxO1, transcription factor; ACE2, angiotensin-converting enzyme 2; mTOR, mammalian target of rapamycin kinase; XIAP, BCL3, BCL2, BOK pro-apoptotic protein; BAX, BAK1, anti-apoptotic protein; NOD1, nucleotide binding oligomerization domain containing 1; CRADD, domain containing adaptor with death domain; APAF1, apoptotic protease activating factor 1.

**Table 2 cells-12-00150-t002:** Pro-apoptotic effects of apelinergic system. Stimulation, ↑.

Tissues	Species	Doses	Times	Signalling Pathways	References
stomach and mid-jejunum	rat	100 nmol/kg/day	for 10 days	↑ caspase-3	[97]
Caco-2	human	10^–8^ M	48 h	↑ caspase-3	[96]
granulosa cells	bovine	-	-	pro-apoptotic effects ↑mRNA APJ expression	[108]
ESCs	mouse	-	-	↑ p53	[107]

Abbreviations: Caco-2 cell line from human colon adenocarcinoma; ESCs embryonic stem cells.

## Data Availability

Not applicable.

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
