# Peer review of "The Apelinergic System: Apelin, ELABELA, and APJ Action on Cell Apoptosis: Anti-Apoptotic or Pro-Apoptotic Effect?"

_cells, 2022, doi:10.3390/cells12010150_

Round 1
Reviewer 1 Report
The authors have submitted a comprehensive review on the apelinergic system and apoptosis, which comes to supplement the many recent reviews on the same system in several pathologies. The manuscript is clear and well written with minor language revisions needed.
I think that the paper would benefit from the following:
1. The authors should add more specific information when they discuss functions of the apelinergic system. E.g. "...a significant impact of apelin on the functioning of the cardiovascular system", or "it is essential for angiogenesis" and so on. In all these cases, they should be more specific as to what does it do/influence to lead to what effect.
Fig.2: It should be preferable to have the same size of letters throughout the figure.
Fig. 3: What is the effect on bone formation via Sox32? Do the effects presented in red come from apelin and those in blue from ELABELA? Clarify in the legend. Do apelin and ELABELA activate the same pathways, or is it not known? Add references for each presented action.
Table 1 should be edited so that it is easily read.
Paragraph 7 (Pro-apoptotic...): Replace the first word Several" by "A few studies..." In the same paragraph the authors say "On the basis of many scientific studies, it has been suggested that apelin impacts the Bax/Bcl-2 ratio [93]" What are the many studies? If there are more, they should be added here.
Conclusions: The authors should comment on the existing literature and give their insight into how the apelinergic system can be exploited therapeutically based on its effect on apoptosis. They should thus comment on whether they believe it is the pro- or the anti-apoptotic function that is major based on the existing literature. What is the basis for the contradictory data and how these may support or oppose the possible use of apelin as a therapeutic target? A schematic would help.
Author Response
The authors have submitted a comprehensive review on the apelinergic system and apoptosis, which comes to supplement the many recent reviews on the same system in several pathologies. The manuscript is clear and well written with minor language revisions needed.
We thank the Reviewer for his/her thorough review and comments. Please find below our responses to concerns regarding the present work. We hope that the revised manuscript will meet the reviewer's expectations.
I think that the paper would benefit from the following:
- The authors should add more specific information when they discuss functions of the apelinergic system. E.g. "...a significant impact of apelin on the functioning of the cardiovascular system", or "it is essential for angiogenesis" and so on. In all these cases, they should be more specific as to what does it do/influence to lead to what effect.
It has been completed as suggested.
Fig.2: It should be preferable to have the same size of letters throughout the figure.
It has been changed as suggested.
Fig. 3: What is the effect on bone formation via Sox32? Do the effects presented in red come from apelin and those in blue from ELABELA? Clarify in the legend. Do apelin and ELABELA activate the same pathways, or is it not known? Add references for each presented action.
Concerning Sox32, I used a mental shortcut, which it has been corrected in the picture - refers to an excerpt from the research included in this manuscript. “Finally, ELABELA/APJ regulates zebrafish skeletal development, bone formation, and bone homeostasis by inhibiting SRY-box transcription factor 32 levels in ventrolateral endodermal cells, indirectly leading to changes the expression of bone morphogenetic protein targets [50].”
The legend has been changed to clarify the colors in the figure.
Yes, it is known: “Both Apelin and ELABELA are a ligand for the APJ receptor, although they have many similarities regarding their functions in the body, they act through different signaling pathways and have different biological effects…”. In additional references have been added in Figure 3.
Table 1 should be edited so that it is easily read.
The table has been divided into appropriate columns, the data has been arranged according to the nervous system, circulatory system, etc. in the same order as in the manuscript. I added information on what arrows are sign the stimulating and inhibiting effect.
Paragraph 7 (Pro-apoptotic...): Replace the first word Several" by "A few studies..." In the same paragraph the authors say "On the basis of many scientific studies, it has been suggested that apelin impacts the Bax/Bcl-2 ratio [93]" What are the many studies? If there are more, they should be added here.
It has been changed as suggested and I added the appropriate references to "On the basis of many scientific studies, it has been suggested that apelin impacts the Bax/Bcl-2 ratio [93]".
Conclusions: The authors should comment on the existing literature and give their insight into how the apelinergic system can be exploited therapeutically based on its effect on apoptosis. They should thus comment on whether they believe it is the pro- or the anti-apoptotic function that is major based on the existing literature. What is the basis for the contradictory data and how these may support or oppose the possible use of apelin as a therapeutic target? A schematic would help.
It has been changed as suggested. New Figure was added as a suggestion.
Reviewer 2 Report
In this well written and quite complete Review article the Authors addressed the role of the apelinergic system in cellular apoptosis. I would suggest mentioning that Apelin role in cancer progression has been reviewed elsewhere, as it is a very relevant and quite extensive topic that has not been fully addressed in the present manuscript. The reference list may also be shortened, selecting a review instead of original papers when appropriate, and updated (e.g. #6, #53) as needed.
Minor points
1. Several acronyms have not been spelled at first use (abstract line 20; par.4 line 209, Table 1 legend, par “reproductive system” line 422)
2. Acronyms use (e.g. Akt or AKT) and British/American spelling (e. g. signalling/signaling) should be standardized.
3. A few references need to be checked as they do not appear to be completely pertinent (e.g.#8,#67)
4. Genes names should be italicized (e.g.APLNR line 191)
5. Please carefully check for typos (particularly in Table 1)
Author Response
In this well written and quite complete Review article the Authors addressed the role of the apelinergic system in cellular apoptosis. I would suggest mentioning that Apelin role in cancer progression has been reviewed elsewhere, as it is a very relevant and quite extensive topic that has not been fully addressed in the present manuscript. The reference list may also be shortened, selecting a review instead of original papers when appropriate, and updated (e.g. #6, #53) as needed.
We thank the Reviewer for his/her thorough review and comments. Please find below our responses to concerns regarding the present work. We hope that the revised manuscript will meet the reviewer's expectations.
It has added information on the role of apelin in cancer progression and therapy and appropriate references to literature sources.
Due to the extensive information on the expression, function, and influence on the apoptosis of the apelinergic system in individual body systems depending on the species, which are additionally summarized in the table, it isn't easy to shorten the references. The vast majority are articles whose exact data are presented in this manuscript.
Minor points
- Several acronyms have not been spelled at first use (abstract line 20; par.4 line 209, Table 1 legend, par “reproductive system” line 422)
It has been changed as suggested.
- Acronyms use (e.g. Akt or AKT) and British/American spelling (e. g. signalling/signaling) should be standardized.
It has been changed as suggested. - A few references need to be checked as they do not appear to be completely pertinent (e.g.#8,#67)
It has been checked and changed.
- Genes names should be italicized (e.g.APLNR line 191)
It has been changed as suggested. - Please carefully check for typos (particularly in Table 1)
It has been changed as suggested.